# HIV Care Profiling and Delivery Status in the Mobile Health Clinics of eThekwini District in KwaZulu Natal, South Africa: A Descriptive Evaluation Study

**Silingene Joyce Ngcobo [1,*], Lufuno Makhado [2]** 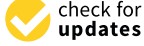 **and Leepile Alfred Sehularo [3]**

1   College Health Sciences, University of KwaZulu Natal, Durban 4041, South Africa
2   Office of the Deputy Dean Research and Postgraduate Studies, Faculty of Health Sciences, University of Venda, Thohoyandou 0950, South Africa
3   NuMIQ Research Focus Area, Faculty of Health Sciences, North-West University, Mafikeng 2531, South Africa
*   Correspondence: ngcobos5@ukzn.ac.za

**Abstract:** Mobile health clinics (MHCs) serve as an alternative HIV care delivery method for the HIV-burdened eThekwini district. This study aimed to describe and profile the HIV care services provided by the MHCs through process evaluation. A descriptive cross-sectional quantitative evaluation study was performed on 137 MHCs using total population sampling. An online data collection method using a validated 50-item researcher-developed instrument was administered to professional nurses who are MHC team leaders, following ethical approval from the local university and departments of health. Descriptive statistics were used to analyze the data. The results described that HIV care services are offered in open spaces (43%), community buildings (37%), solid built buildings called health posts (15%), vehicles (9%), and tents (2%) with no electricity (77%), water (55%), and sanitation (64%). Adults (97%) are the main recipients of HIV care in MHCs (90%) offering antiretroviral therapy (95%). Staff, monitoring, and retaining care challenges were noted, with good linkage (91%) and referral pathways ($n$ = 123.90%). In conclusion, the standardization and prioritization of HIV care with specific contextual practice guidelines are vital.

**Keywords:** mobile health clinics; cross-sectional descriptive evaluation study; HIV care; mobile health clinic profiling; HIV delivery method

## 1. Introduction

Currently, it is reported that approximately 39 million people in the world are living with HIV, and 8.5 million (20%) live in South Africa (SA) [1]. Contextually, this is a significant public health concern [2], especially since the country has also reported an increase in HIV incidence [3]. Despite having the largest antiretroviral therapy (ART) rollout program globally [4], evidence in the country suggests that not all people living with HIV (PLHIV) needing HIV care are actually receiving it, due to various challenges [5,6]. This dampens HIV care and support services' desire to reach universal coverage [7]. However, the offer of extended HIV care through various modalities such as mobile health clinics (MHCs) in SA proves to improve and increase access to HIV care services. MHCs significantly serve as one of the strategic interventions towards addressing the quadruple burden of the diseases encompassing the country's current public health system [8], which include HIV. In this context, MHCs are being embraced as an additional healthcare delivery modality aimed at HIV program expansion, without misplacing their initial inception purpose of serving general populations with healthcare access challenges [9,10]. MHCs are an officially acknowledged and accepted primary health care (PHC) service extension for remote areas within the local district health system [10].

Prevailing community health needs inform primary health care service package that is offered in the MHCs, which is often similar to what is available in community PHC

clinics. However, various factors, including but not limited to feasibility, human resources, and the context of the MHCs, eventually inform the type of service being offered, which should also include HIV care. Therefore, the HIV care services in MHCs critically need evaluation for the purpose of determining the efficiency of HIV care management in such settings. An evaluation is an organized assessment undertaken to determine the features, facilities, procedures, and results of programs [11]. Furthermore, real-life situations are reflected through evaluation actions [12], and any healthcare service evaluation is essential, because it provides valuable feedback [8]. The literature outlines various types [13,14] and reasons [12,15,16] for conducting an evaluation. This study adopted a research-centered evaluation approach by the means of process evaluation, because it allows for good generalizability [12]. Furthermore, process evaluation focuses on examining how something happens, as opposed to measuring its outcomes [12,17,18]. According to Naidoo [19], an assessment of performance and inspection of internal and external activities to determine if the efforts match the effects and results within programs or organizations are part of the three-dimensional all-inclusiveness accepted in evaluations. The value of process evaluation includes making informed future choices, efficiency improvements, and making intended decisions about processes and activities [11]. This evaluation study provides a synopsis reflection of the current HIV care and staffing profile in the MHCs of eThekwini district, in the presence of anecdotal evidence regarding the HIV care services provided by MHCs nationally. The objectives of the study were to describe and profile MHCs regarding the staffing and HIV care provision status in the MHCs of eThekwini District. The eThekwini district is the epicenter of the HIV epidemic [20] in KwaZulu Natal (KZN) province, which has the highest national HIV prevalence rate at 18.9%, especially amongst 15–19 year olds [21].

## 2. Methods

### 2.1. Study Design

A descriptive, cross-sectional, quantitative evaluation study design was adopted for the MHCs within the eThekwini municipality. Descriptive, cross-sectional evaluation designs provide statistics for describing the condition of phenomena or associations among phenomena in a given time period; furthermore, they are helpful for public health planning, monitoring, and evaluation [22]. The profiling of 137 MHCs and descriptions of their HIV care service provisions were achieved in the study, constituting accurate accounts of the characteristics of particular individuals (staff complements), situations (conditions), and services [23].

### 2.2. Setting

eThekwini is one of the biggest metros in the province of KZN, which had a population of approximately 3,199,000 in June 2022 [3]. It has a combination of urban, peri-urban, and rural communities which receive healthcare services through a district health system (DHS) approach [24]. Both provincial and local governments are responsible for health care services' provision at various levels. The former contributes 60% and the latter 40% of those health services [25–27], which include MHC modalities. The majority of the population rely on public health services, since gross disparities, poverty, and unemployment are major challenges present within the district [28]. These challenges have a direct effect on health. A total of 36 MHC teams were found within the district and each MHC team services multiple locations (between 5 and 20) referred to mobile points. These mobile points constitute a functional standalone MHC and vast differences exist from one MHC to the next.

### 2.3. Study Population and Sampling Strategy

A total population sampling (TPS) technique was used for this evaluation study, because TPS is best suited for adoption if the total population size is less than 200 [8,29]. Such an adoption aids in the elimination of bias and reveals a phenomenon in its real state [29]. In total, 137 MHCs were included in the study, which are operated by both the local and provincial departments of health. Other MHCs operated by private, non-

government, and non-profit organizations were excluded, since their service sustainability depends on donor funding availability.

### 2.4. Data Collection

A 'socially distant' data collection method [30] through the means of WhatsApp was used, as the study was conducted during the COVID-19 pandemic, when national lockdown restrictions were implemented. Therefore, the researchers had restricted physical contact with the MHCs' staff, in order to comply with health and safety restrictions. After obtaining gatekeeper and department of health permissions for the study, human resources (HR) departments provided the contact details of all nurse leaders. A process of obtaining informed consent was undertaken by an independent person ensuring that three goals of informed consent were covered thoroughly, i.e., information disclosure, comprehension of the information, and voluntariness [31] were covered. A Google form which contained the study information and questionnaire was sent via WhatsApp to all the MHCs' nurse team leaders. The link had an 'I agree to participate' button, which, if clicked, then gave consent to participate in the study. The administered Google form questionnaire had two choices for each question, which the respondent chose their choice from. Diverse information was collected through the Google forms in a simple and effective way, employing different types of questions [32]. There was an 89% response rate from 32 MHC teams representing 137 MHCs, while 4 MHC teams did not participate. The four MHCs which did not participate have five MHCs each attached to them. Therefore, 20 MHCs were not included in this evaluation. Online surveys are suitable for evaluation studies, including descriptive, case–controls, and cohort studies [33].

### 2.5. Data Analysis

Descriptive statistics were used to analyze 50 quantitative, close-ended, structured questions addressing the MHCs' profiling, staffing, and HIV care provision on a dichotomous scale. Measures of dispersion, particularly the frequency distribution of all the study variables, were descriptively analyzed. Prior to analysis, using SPSS version 27, the data were cleaned and entered into an excel spreadsheet.

### 2.6. Research Instrument

An online, researcher-developed research instrument was used. Two source documents were used for the instrument development, which were the PHC supervisor's manual [34] and the United States President's Emergency Plan for AIDS Relief (PEPFAR)'s framework on HIV support and care. Additionally, concepts contained in the structural components of Donabedian's Structure-Process-Outcome (SPO) model [35] guided the formation of the questions. Donabedian's SPO study was the main conceptual framework that underpinned this evaluation study. A total of 50 questions made up the instrument, which had dichotomous variables requiring "yes" or "no" responses. A few questions required rationale and were left open-ended. The tool specifically focused on questions relating to profiling, infrastructure and municipal services, HIV care support and NIMART services, HIV policies and guidelines, equipment, records and information systems, supervision, and safety in mobile health clinics.

### 2.7. Validity and Reliability of the Instrument

Three ways were employed to ensure content validity for the study instrument. Firstly, the concepts used in the questionnaire were extracted from the SPO model [35]; secondly, certain contents of the tool were directly from the PHC supervisory manual, which has been in use in South Africa's PHC clinics for over a decade [36]; and thirdly, the study's principal investigator was consulted as an expert [37] on the subject at hand. Reliability is ensured when the research instrument produces the same results on repeated occasions, i.e., the consistency or stability of scores over time or across ratters [38]. In this study, test–retest reliability was conducted on the study instrument two weeks apart, and the

Persons correlation coefficient value was 0.716. Furthermore, a Cronbach's alpha score of 0.865 was calculated to measure the internal consistency of the instrument; the literature [39] confirmed that such a score is good.

*2.8. Ethical Consideration*

Permission to conduct the study was granted by the local University Health Research Ethics Committee, reference number (Ethics Number: NWU 00934-19-A1), the provincial department of health, ref. no (KZ_202002_017), and the local municipality health department, ref. no (30/1/1/6/3/1). Furthermore, gatekeepers' permission was obtained from the district office and chief executive officers of various community health centers which have MHCs and MHC managers. All domains of voluntariness were observed during the study, as outlined by [40]. Confidentiality [41] and anonymity [42] were ensured, as no personal information was required and clinic names were de-identified. A participant information leaflet to ensure informed consent [43] was supplied.

### 3. Results

*3.1. Profiling of MHCs*

A total of (*n* = 137) MHCs which are operated by both local (24%) and provincial (76%) health departments in eThekwini district were subjected to an evaluation. The majority (58%) open once a month and offer health services in various settings, including open spaces (43%), community buildings such as halls, churches, homes, and creches (37%), solid buildings purposed for MHCs visits referred to as health posts (15%), vehicles (9%), and tents (2%). More than one combination of settings is used in most cases. The majority reported having an unavailability of basic municipal services such as electricity (77%), water (55%), and sanitation (64%). Only (0.7%) have a reliable means of communication, and (99%) have no patient transport available in an emergency.

Fifty one percent of the MHCs were located more than 21 km away from the central departure point and spent up to 120 min traveling time per single trip. The majority (49%) spent up to 30 min, while 46% spent from above 30 min to an hour and 5% spent up to two hours. However, 58% reported good road conditions consisting of tarred (74%) and 26% gravel roads. Table 1 depicts all the profiling variables measured in the study, while Table 2 outlines the nature of the settings/approaches that the health services are offered in the MHCs.

**Table 1.** Profile of eThekwini district mobile health clinics.

| Variables | Attributes | Freq (%) *n* = 137 |
|---|---|---|
| MHC operated by: | Municipality | 24 |
| | Government | 76 |
| MHC operating frequency per month: | Once | 58 |
| | Twice | 20 |
| | 3–4 times | 1 |
| | 5 or more times | 21 |
| Distance between central point and MHCs | Within 5 km | 11 |
| | 6–10 km | 23 |
| | 11–20 km | 15 |
| | 21 km and above | 51 |
| One-way travel time it takes to reach MHC location. | 0–30 min | 49 |
| | 31–60 min | 39 |
| | 61–90 min | 7 |
| | 91–120 min | 5 |
| Terrain conditions to the mobile health clinic: | | |
| Good | yes | 58 |
| Gravel | yes | 29 |
| Tarred | yes | 75 |

**Table 1.** *Cont.*

| Variables | Attributes | Freq (%) *n* = 137 |
|---|---|---|
| Availability of essential services: | | |
| Electricity | yes | 23 |
| Proper sanitation | yes | 37 |
| Water supply | yes | 45 |
| Reliable emergency transport | yes | 1 |
| Telephone or a two-way radio | yes | 1 |

**Table 2.** Healthcare delivery approaches through MHCs in eThekwini district.

| Type of a Setting for MHC | Frequency (%) *n* = 137 |
|---|---|
| a.  Vehicle | 9 |
| b.  Health post (solid building purposefully built to be used as an MHC in the community) | 15 |
| c.  Community building, e.g., church, hall, creche, and home, etc. | 37 |
| d.  Tent | 3 |
| e.  Open space (outside) | 36 |

Staffing: Different staff compositions and complements were found in the MHCs and consisted of various members, i.e., nurses (professional, enrolled, and enrolled nursing axillary), HIV counsellors, drivers, cleaners, and security guards, who are outlined in Tables 3 and 4.

**Table 3.** Staff complements at MHCs and those offering HIV care services.

| Variables | Attributes | Frequency % *n* = 137 |
|---|---|---|
| Total number of personnel that form part of this mobile health clinic team | 1–3 | 13 |
| | 4–6 | 77 |
| | 7–9 | 10 |
| Number of professional nurses are working/allocated mobile health clinic | 1.00 | 15 |
| | 2.00 | 76 |
| | 3.00 | 10 |
| Number of NIMART-trained nurses in the MHC | 1.00 | 60 |
| | 2.00 | 40 |
| Categories of staff that offer HIV care and support services in mobile health clinic | Professional nurse and counsellor | 3 |
| | Professional nurse | 2 |
| | Professional nurse and enrolled nurse | 87 |
| | Professional nurse, enrolled nurse, and counsellors | 8 |

**Table 4.** Support staff available at MHCs.

| Titles of the Other Personnel Working in Each MHC | Frequency (%) *n* = 137 |
|---|---|
| a.  Enrolled nurse | 12 |
| b.  Enrolled nurse, enrolled nursing auxiliary, and HIV counsellor | 39 |
| c.  Enrolled nurse, enrolled nursing auxiliary, and clerk | 3 |
| d.  Enrolled nurse, enrolled nursing auxiliary, and security | 16 |
| e.  Enrolled nurse and HIV counsellor | 19 |
| f.  Enrolled nurse, enrolled nursing auxiliary, clerk, and security | 1 |
| g.  Enrolled nursing auxiliary and HIV counsellor | 1 |
| h.  Enrolled nursing auxiliary, 2 HIV counsellors, and clerk | 8 |
| i.  Enrolled nursing auxiliary, HIV counsellors, and clerk | 1 |

*3.2. HIV Care*

Ninety percent of the MHCs in eThekwini do offer certain HIV care services to the communities that they visit. The recipients (97%) of HIV care are mostly the adult population,

and ART is provided by 95% of MHCs for treatment purposes. Meanwhile, 100% MHCs offer HIV testing and counselling and TB screening. Patient monitoring and retaining their care were noted as challenges faced by MHCs, even though 91% reported a good linkage to care, with 90% having clear referral pathways available. Table 5 depicts the various HIV care and support services available in MHCs.

**Table 5.** HIV care services within the MHCs of eThekwini metro.

| Items | Attributes | Freq (%) |
|---|---|---|
| Does this mobile health clinic offer any HIV care and support services? | yes<br>no | 123 (89.8)<br>14 (10.2) |
| Is there a linkage to the care of HIV-infected patients available in this mobile health clinic? | yes<br>no | 125 (91.2)<br>12 (8.8) |
| Are HIV care and support services offered to adults only? | yes<br>no | 133 (97.1)<br>4 (2.9) |
| Are HIV care and support services offered to children only? | yes<br>no | 5 (3.6)<br>132 (96.4) |
| Is HIV Counselling and Testing (HCT) offered at this mobile health clinic? | yes | 137 (100.00) |
| Does this mobile clinic have antiretroviral therapy (ART) available for HIV? | yes<br>no | 130 (94.9)<br>7 (5.1) |
| Is ART available for prevention purposes? | yes<br>no | 9 (6.6)<br>128 (93.4) |
| Is ART available for treatment purposes? | yes<br>no | 130 (94.9)<br>7 (5.1) |
| Does this mobile health clinic initiate patients on ART (HAART)? | yes<br>no | 130 (94.9)<br>7 (5.1) |
| Does this mobile health clinic retain patients on HAART? | yes<br>no | 69 (50.4)<br>68 (49.6) |
| Does this mobile health clinic receive down referral patients for ART? | yes<br>no | 50 (36.5)<br>87 (63.5) |
| Does this mobile health clinic perform clinical monitoring of HIV-diagnosed patients? | yes<br>no | 88 (64.2)<br>49 (35.8) |
| Does this mobile health clinic do laboratory monitoring for HIV-diagnosed patients? | yes<br>no | 41 (29.9)<br>96 (70.1) |
| Is equipment for collecting blood and samples available at this MHC? | yes<br>no | 55 (40.1)<br>82 (59.9) |
| Does this mobile health clinic do CD4+ T cell counts? | yes<br>no | 47 (34.3)<br>90 (65.7) |
| Does this mobile health clinic do HIV viral loads? | yes<br>no | 28 (20.4)<br>109 (79.6) |
| Is there a clear referral pathway available at the mobile health clinic for HIV-infected patients? | yes<br>no | 123 (89.8)<br>14 (10.2) |
| Does this mobile health clinic offer tuberculosis screening? | yes | 137 (100.0) |
| Does this mobile clinic offer cotrimoxazole prophylaxis? | yes<br>no | 124 (90.5)<br>13 (9.5) |

## 4. Discussion

The MHCs were staffed with various members who fulfilled various roles, as shown in Tables 3 and 4, and professional (registered) nurses (RN) were the most qualified and senior members leading the MHC teams. In total, 87% of HIV care and support responsibilities were offered by registered and enrolled nurses. That is an acceptable norm in

the context, as nurses are the heartbeat of MHC operations [25], driving decentralized HIV care programs [44] while being supported by task-shifting or task-sharing [45] endorsements. Furthermore, nurse-initiated management antiretroviral therapy (NIMART) training and practice support that nurses should render the relevant HIV care in PHC clinics independently. NIMART consists of HIV diagnosis and clinical staging, prescribing first- and second-line ART, and managing common opportunistic infections and other treatment-related conditions [46].

Between 2010 and 2016, eThekwini district trained 904 nurses on NIMART [47], and this training continued beyond 2016, as departments of health and various training institutions still offer the training. This study revealed that each MHC clinic within the district had at least one NIMART-trained RN, while 40% of them had two NIMART-trained RNs. This meant that the nurses in the MHCs were equipped to offer much-needed competent HIV care and support services, even though inconsistencies in staff allocation were observed throughout the MHCs, evidenced by some MHCs having between one and three RNs allocated to them. In total, 76% of the MHCs had two RNs and, seldom, 10% had three RNs. These inconsistencies in work allocation could impact on the workload distribution, staff coping mechanisms, and quality of care provided, because it this not in line with the current recommended ratio of 4.45 per 1000 population in healthcare [6,48]. Yet, the clinical consultation of PLHIV is unpredictable depending on the presenting need, resulting in an inability to precisely measure the time spent on consultation for each patient. A total of 90% of MHCs in the eThekwini district offer certain HIV care and support services, as outlined in Table 5, which translates into the district's population having somewhat increased HIV care access due to MHC presence. However, it has been reported in the literature [6,49,50] that increasing HIV access does not automatically mean or result in care and treatment without interruptions over time. Therefore, a need to examine the type of HIV care and support services offered in the MHCs is critical, including sustainability measures.

This study demonstrated that 100% of the MHCs provided HIV counselling and testing, which is similar to other sub-Saharan African study findings, whereby community-based testing reported a better and increased uptake compared to facility-based testing [51–53]. This was commendable for the MHCs, because it confirms the successful expansion and supplementation of existing HIV testing from fixed health facilities while attaining increasing HIV testing. This was not an unexpected finding, since MHCs are known to serve and attract hard-to-obtain individuals through community-based HIV testing [54–56].

MHCs testing services are central to increasing awareness of HIV status [57], and no HIV testing takes place without counselling. Both are critical initial steps toward HIV prevention and treatment [52]. Therefore, the services of an HIV counselor would be highly beneficial in all MHCs. However, approximately 60% of the MHCs did not have counsellors in the study, meaning that additional load was added on nurses, who should dedicate time to offer counselling, separate to clinical services.

Visibly, 97% the HIV care services within the MHCs were delivered primarily to adult patients, with a 91% reported good linkage to care. These findings were higher than those reported in a Tanzanian study [58] with similar settings. However, only 4% of the MHCs provided pediatric HIV care. This was not surprising, as a general lack of confidence in nurses providing pediatric HIV care despite having received NIMART training has been previously documented in the literature, including in different parts of South Africa [59,60], Uganda [61], and Tanzania [62]. Furthermore, other researchers [47,63,64] have also reported that having nurses trained in NIMART does not necessarily mean that these nurses, post-completion of the course, are confidently able to implement what they have been taught. However, in this study, nurses from 93% MHCs reported staff and equipment shortages as the main reasons for not providing pediatric HIV as opposed to only lacking confidence. The eThekwini municipality had a vacancy rate of 15.4% as of 31 March 2022 [65]. Therefore, HIV care clinical mentorship and support implementation programs in MHCs are critical and should contribute towards improving pediatric HIV services uptake, while also providing a sense of support to MHC nurses to not feel isolated and alone. The 91%

good linkage to care reported by the MHCs is somewhat encouraging that pediatric HIV care cases are being referred and linked to care, despite having no tracing mechanisms reported to be in place for all the referred and linked cases. Therefore, advocacy for the tracing of all referred cases should be implemented and accurately articulated. Such implementation will ensure that pediatric cases from MHCs receive appropriate care, even if such care is not provided directly by MHCs. Ultimately, MHCs can be deemed as effective for the delivery of HIV care for both infected adult and children populations.

In the context where this study was conducted, HIV still remains the major public health concern [66]; therefore, MHCs add a positive stride towards the current implemented HIV measures directed at strategies for the control and management of the epidemic. If the knowledge of HIV status increases [67], it will ultimately increase the uptake of ART initiation for newly diagnosed PLHIV [68–70]. In total, the 95% ART availability in the study was motivating to note, even though it was only available for treatment purposes, but not for those needing it for prevention purposes. Also, these findings denoted that, out of 100% testing HIV positive, 5% of those would not have the necessary HIV care needed in terms of treatment initiation. It is hoped that that these 5% benefit from the reported good linkage to care and referral. Strengthening the HIV care services in MHCs is desired in order to ensure that all receive care, as stipulated by policies.

Undoubtedly, MHCs can be seen as one of the effective methods in HIV care delivery, because some of the findings in this study are significantly higher than what is widely accepted and reported in the literature. In respect to an expected drop-off rate along the HIV continuum [71–73], as depicted in the recent 2020 UNAIDS 90-90-90 goals attained generally, where a decline was observed at each stage [74], the provision of and making ART available are both human rights and biomedical obligations, since its administration leads to HIV transmission decline at the individual and population levels [75–77]. Therefore, ART availability within MHCs improves local communities' health outcomes by controlling transmission while simultaneously eliminating, to a certain extent, multiple barriers previously reported, such as inaccessibility and navigating through the health system [73,78]. MHCs provide access to the healthcare system for the local population through their backyards, and community-based ART availability and administration have been reported to yield favorable outcomes, including good treatment adherence [79], while, on the contrary, the non-provision of ART leads to excess morbidity, mortality, and viral transmission [80].

Therefore, the 5% of MHCs that were reported to not render ART cited that ART in 40% of MHCs was only made available to pregnant women during their initial visit. In total, 60% reported that there was lack of continuity of care, since they visit MHCs just once a month and cannot perform appropriate follow-ups, especially for the patients that they will be initiating on treatment. Therefore, they opt to send those patients to fixed clinic facilities. Hence, the MHCs reported a 50% ability to retain patients in care in their capacity. In both instances, they do provide ART during that initial visit and then refer to the fixed clinics for follow-up care. Therefore, they regard themselves as not providing the service. This is of concern in light of already reported increased losses in follow-up and attrition rates from HIV care services following diagnosis [81,82]. A poor or lack of continuity of care will further perpetuate a rise in attrition rates, since inadequate and poorly understood linkages to HIV care following MHC services have already been reported in the literature [83–85].

Nevertheless, MHCs' ART availability is an important health indicator. It relates to the implementation of core activities relating to essential medicines and ART access [86], including scaling up ART in the public sector [2]. It is motivating to see such implementation within the MHCs' context, informed by public health and universal health coverage approaches. However, 95% of the MHCs had ART only for treatment purposes, while only 7% had it accessible as a preventative therapy. The reason for low PrEP availability is the fact that, within the country, only certain populations are regarded as eligible for PrEP and, in the MHCs, the focus is mainly the accessibility of ART for those with a positive HIV diagnosis. This communicates an inequitable access of ART in the MHCs, since ART is the foundation for both HIV prevention and management [87]. Therefore, it calls for

policy makers and implementers to attend to this identified gap and capacitate staff and community accordingly in order to rollout ARVs as a preventative modality. This study revealed that MHCs in the districts have a partial availability of ART.

Further challenges were noted regarding the monitoring of PLHIV and retaining them in care while in the MHCs, because only 50% of the patients were retained in care. This raises concerns regarding the available follow-up care in the MHCs and no evidence in the literature exploring the follow-up care of ART patients from MHCs has been found. Patient follow-up investigations after ART initiation from MHCs are highly recommended. As the situation stands currently, MHCs can be interpreted as failing to meet community HIV needs somewhat due to the partiality of HIV services rendered there. Yet, MHCs are known to reach mostly socioeconomically challenged communities with health service needs; unfortunately, the current status quo compels for patients to be referred to a fixed facility, which was thought to be inaccessible initially. In hindsight, MHCs have addressed some barriers through their ability reach communities despite conditions, time, and traveling distance factors as depicted in Table 1.

Furthermore, only 37% of MHCs accepted down referrals for HIV-related cases, meaning that continuity of care is inadequate to a certain degree and also denotes that proper communication between MHCs and other fixed health institutions is not fully functional. Lack of communication could be contributed by the fact that less than 1% of the MHCs had communication tools available, and also the limited numbers of HIV counsellors across MHCs attributed to poor referral systems. The value and function of HIV counsellors goes beyond just counselling a patient in front of them; they play a big role in tracing and following up on referrals [88] at various intervals. Several researchers [81,89] have affirmed the above cited reasons. Furthermore, a combination of social, structural, and psychological patient factors and even poor health care infrastructure [90] contribute towards a lack of continuity for care. A careful and well thought out strategic plan enabling the effective implementation and sustainability of HIV care in MHCs is vital in order to eliminate the partial offering of HIV care services. Alternative ways of delivering care can be considered, such as a hybrid approach to treatment, which has demonstrated favorable results in the past [91]. This will bring about the standardization of HIV care across all MHCs, since 90% of the MHCs reported having clear referral pathways. Studies in eastern Africa have concluded that streamlined ART delivery, facilitated linkage, and recurrent support through counsellors lead to success in community-based HIV care programs, even though additional resources are needed to support the stated interventions [88,92].

This evaluation study further provided evidence on another existing challenge faced by MHCs pertaining to the monitoring of PLHIV. Only 30% of the MHCs had the capacity to perform mandatory monitoring and 40% had the appropriate equipment to perform the tests. The monitoring tests performed were at minimal ratios of 34% for CD4+ T cell count and 20% for viral loads. The use of point-of-care tests could resolve this issue because of their proven effectiveness in similar settings to MHCs [88].

## 5. Conclusions

The role that MHCs are playing in the plight against the HIV epidemic in eThekwini does contribute towards positive health outcomes, even though multiple challenges are present. The strengthening and support of HIV care programs delivered through MHCs will endeavor an additional positive trail towards ensuring universal health coverage realization. Through the standardization and coordination of MHCs' activities, effective HIV care is possible.

## 6. Recommendations

Considering these current evaluation findings, the following recommendations are made in order for HIV care and support services in MHCs to function optimally.

*6.1. Human Resource Management*

Staffing needs: sufficient staffing needs need to be provided for MHCs and should ensure adequate coverage in terms of all essential categories of the necessary staff for HIV services, i.e., a minimum of two professional nurses, one enrolled nurse, one HIV counsellor, enrolled nursing auxiliary, and support staff. This recommendation will ensure that the current district population of 1741/km$^2$ receive appropriate care [65].

*6.2. Capacity Building and Support*

The development of an HIV mentorship program which will provide support and be available for nurses in the MHCS is crucial. The program will serve as a knowledge hub, thereby assisting in ensuring that nurses gain confidence in the way they render care, which will broaden their scope in terms of the population that they serve to include pediatric HIV patients. Ultimately, comprehensive HIV care package services would be made available. Communities of best practice in the HIV care of MHCs can be developed through mentorship programs, as networks could be formed with other members of healthcare teams, leading to strong partnerships and collaborative practice. Within the mentorship program, various members of the healthcare team should be available.

*6.3. Operations*

Careful planning and investment should be made towards MHCs in order for standardized HIV care to be achieved throughout all MHCs. Clear directives should be developed addressing the infrastructure improvements to the current settings of MHCs depicted in Table 2 and feasible linkage to care patterns to and from the MHCs, while also paying attention to point of care monitoring, CD4+ T cell counts, hemoglobin, creatinine, and viral loads test provision and availability.

*6.4. Further Research Need*

Further studies could be conducted in various districts and provinces to engage the general status of MHCs in the country with respect to HIV care provision, engaging different research methodologies. The research findings will inform policy formulations directly tailored for MHCs settings and would be made to meet both the community needs and staff needs. Also, this descriptive study aimed at profiling the services available without evaluating the outcomes achieved as a result of the study. Therefore, other forms of study can be conducted, including impact studies and qualitative studies focusing on the recipients of MHCs' HIV care and providers, as well including their managers.

**7. Study Limitations**

This study was only conducted in one semi-rural but wealthiest district of KwaZulu Natal province. Therefore, the findings cannot be generalized to the rest of the province nor the country. Also, the impact of the HIV care provided by the MHCs was not measured, and only the feasibility, availability, and nature of where MHC services were provided were the focus of the evaluation.

**Author Contributions:** Conceptualization, S.J.N. and L.M.; methodology, S.J.N.; software, L.M.; validation, S.J.N., L.M. and L.A.S.; formal analysis, L.M.; investigation, S.J.N.; data curation, L.A.S.; writing—original draft preparation, S.J.N.; writing—review and editing, S.J.N., L.M. and L.A.S.; supervision, L.M. and L.A.S. All authors have read and agreed to the published version of the manuscript.

**Funding:** This research received no external funding.

**Institutional Review Board Statement:** The study was conducted in accordance with the Declaration of Helsinki and approved by the Institutional Review Board (or Ethics Committee) of North-West University (Ethics number NWU 00934-19-A1 and date of approval was 11 April 2021).

**Informed Consent Statement:** Informed consent was obtained from all subjects involved in the study.

**Data Availability Statement:** The data presented in this study are available on request from the corresponding author. The data are not publicly available due to the nature of ethical approval.

**Public Involvement Statement:** Members of the public and patients from remote areas played a crucial role in shaping this research. Individuals with a keen interest in the study's subject matter were invited to provide valuable insights during the study design phase and offered feedback on research materials. Their contributions significantly enhanced the relevance and ethical considerations of this work. We extend our sincere appreciation to these individuals for their enthusiastic participation.

**Guidelines and Standards Statement:** This research adheres to the applicable guidelines and standards relevant to the field. We have followed STROBE for observational studies to ensure the rigor and transparency of our methodology. Ethical considerations were addressed in accordance with North West University Health Research Ethics Committee, KwaZulu Natal Provincial Health Department and eThekwini Municipality Local Health Department ethics boards We are committed to upholding the highest standards of research integrity and compliance with established protocols.

**Conflicts of Interest:** The authors declare no conflict of interest.

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
