# Peer review of "HIV Care Profiling and Delivery Status in the Mobile Health Clinics of eThekwini District in KwaZulu Natal, South Africa: A Descriptive Evaluation Study"

_nursrep, doi:10.3390/nursrep13040129_

Round 1

Reviewer 1 Report

Comments and Suggestions for Authors

The manuscript, “HIV care profiling and delivery status in the Mobile Health 2 Clinics of eThekwini District”, characterizes mobile HIV care clinics in an area of South Africa with a high burden of HIV infection. The findings are interesting and relevant to understanding the epidemiology of the region. While there is merit to the manuscript, it requires significant editing to read more concisely and clearly. The discussion could also benefit from better organization for easier reading and understanding the contextualization of findings. Additionally, the authors are encouraged to temper some of their discussion statements that are not fully supported by findings. Other suggestions to improve the manuscript are provided below.

The introduction and all other parts of the manuscript (including tables) can benefit from closer attention to grammar and spelling; further editing is required. For example, line 42 - “...and have proved to substantially benefiting PLHIV...”; line 51 - “HIV care inclusion in MHCs is seen as one of an important interventions...”; line 52 - “quadrable...”

How many MHC were represented by the 4 teams that did not participate?

The authors do not present any data describing the numbers or volumes of people serviced by these MHCs. Given this, it is difficult to ascertain the needs of staff or the community in a way that corroborates statements made in lines 229-233.

The authors also overstate their findings in other ways. For instance, in lines 284 onwards, they state that there is “100% availability... for those who had tested HIV positive” and “This is a positive finding because it means that no patient needing ART was missed.” Such conclusions cannot be drawn from the result that 95% of MHCs provide ART. 

Lines 287-289: How are you able to compare the drop-off from findings in this study to determine that levels are significantly higher? From the results, 95% of MHCs provide ART but only 50% retain patients on ART.

Could a reason for the markedly low rates of ART for HIV prevention be because the question did not ask about PrEP or PEP use? Many people may not consider PrEP or PEP to be ART for prevention.

Reviewer 2 Report

Comments and Suggestions for Authors

Summary

This original study provided a descriptive evaluation of HIV care profiling and delivery in mobile health clinics in South Africa. It is interesting and important to the field of HIV care continuum and can be helpful for future work. Overall I think this original study had an ok quality. Despite the importance of qualitative reports, finer analysis is still warranted to improve the importance of work, especially given the small sample size. Hence, I do not think this report would meet the publication standard but should remain as a community report. Additionally to the public health importance and small sample size concern, I still have some comments, which I believe would be helpful to the authors:

Introduction

1.     Please check typos and grammar in the introduction and throughout the manuscript

2.     Also, it will be helpful for the authors to include a clearly stated study objective in the introduction.

Methods

1.     When reporting reliability, please included Cronbach’s alpha instead of only stating: “Reliability is ensured when the research instrument produces the same results on repeated occasions i.e. consistency or stability of scores over time or across ratters”

2.     Please also provide the 95%CI for Cronbach’s alpha

3.     Clearly, the authors should include the reference for their ethical statement…

.

Results

1.     In the profiling of MHC section, sometimes the authors used comma or semi-comma to separate n and %, it is very confusing and hard to read.

Minor Issues

1.     Can the authors use Vancouver style with [] for reference citing?

2.     Covid-19 should be capitalised

Reviewer 3 Report

Comments and Suggestions for Authors

The authors present a thoughtful manuscript on a prominent health concern in South Africa.  Great effort was made to provide detailed descriptions and justify the selected setting and research design. However, minor grammatical errors in the manuscript distract from the overall message. Additionally, the manuscript is quite dense and could be condensed a bit to focus more on the key messages. 
